# Chemerin Levels in Individuals with Type 2 Diabetes and a Normal Weight versus Individuals with Type 2 Diabetes and Obesity: An Observational, Cross-Sectional Study

**DOI:** 10.3390/biomedicines12050983

**Published:** 2024-04-30

**Authors:** Aishee B. Mukherji, Victoria Idowu, Lei Zhao, Lawrence L. K. Leung, Sa Shen, Latha Palaniappan, John Morser

**Affiliations:** 1Division of Primary Care and Population Health, Stanford University School of Medicine, Stanford, CA 94305, USA; aisheem@stanford.edu; 2Division of Endocrinology, Gerontology and Metabolism, Stanford University School of Medicine, Stanford, CA 94305, USA; vidowu22@siumed.edu; 3Division of Hematology, Stanford University School of Medicine, Stanford, CA 94305, USA; lzhao02@stanford.edu (L.Z.); lawrence.leung@stanford.edu (L.L.K.L.); 4Veterans Affairs Palo Alto Health Care System, 3801 Miranda Avenue, Palo Alto, CA 94304, USA; 5Quantitative Sciences Unit, Stanford University School of Medicine, Stanford, CA 94305, USA; sashen@stanford.edu; 6Division of General Medical Disciplines, Department of Medicine, Stanford University School of Medicine, Stanford, CA 94305, USA

**Keywords:** adipokine, chemerin, diabetes, inflammation, insulin resistance, obesity, protease, type 2 diabetes

## Abstract

Chemerin acts as both a chemotactic agent and an adipokine that undergoes proteolytic cleavage, converting inactive precursors into their active forms before being subsequently inactivated. Elevated chemerin levels are linked to obesity and type 2 diabetes mellitus (T2D). This study aimed to elucidate the effects of T2D and obesity on chemerin levels by comparing plasma samples from individuals with a normal weight and T2D (BMI < 25; NWD group *n* = 22) with those from individuals who are overweight or obese and have T2D (BMI ≥ 25; OWD group *n* = 39). The total chemerin levels were similar in the NWD and OWD groups, suggesting that T2D may equalize the chemerin levels irrespective of obesity status. The cleavage of chemerin has been previously linked to myocardial infarction and stroke in NWD, with potential implications for inflammation and mortality. OWD plasma exhibited lower levels of cleaved chemerin than the NWD group, suggesting less inflammation in the OWD group. Here, we showed that the interaction between obesity and T2D leads to an equalization in the total chemerin levels. The cleaved chemerin levels and the associated inflammatory state, however, differ significantly, underscoring the complex relationship between chemerin, T2D, and obesity.

## 1. Introduction

Chemerin, also known as retinoic acid receptor responder protein 2 (RARRES2) or TIG2, is an adipokine and immunomodulator encoded by the RARRES2 gene [1,2,3,4,5,6]. This adipokine is synthesized as an inactive precursor that undergoes activation and a subsequent inactivation through a series of proteolytic cleavages [7,8]. The inactive circulating precursor, chemerin163S (chem163S) in humans, is converted into a form called chem158K, which possesses 5% of fully active chemerin [9]. Subsequently, it undergoes further cleavage to form chem157S and chem156F, both of which are fully active [10]. The loss of another amino acid results in the formation of inactive chem155A, and further degradation can occur [11]. This mechanism of activation and inactivation through proteolysis has been conserved, as these C-terminal amino acid sequences are highly conserved in all mammals, and homologous forms of chemerin have been identified in mice [12].

There are two signaling receptors for chemerin—chem1 (CMKLR1) and chem2 (GPR1)—plus a receptor that binds to chemerin, CCRL2, presenting chemerin to chem1 or chem2 [13]. Chem157S is fully active on both chem1 and chem2, but differences in their responses to chem156F may exist [13,14]. Chem1 and chem2 differ in their intracellular signaling as well and are differentially expressed in different tissues [7,13], resulting in differences in their physiological and pathophysiological roles.

Chemerin acts as a chemoattractant for some immune cells [15] and is essential for the differentiation and function of white and brown adipose tissues [4,16,17]. In healthy individuals, most of the circulating chemerin is secreted by the liver and exists as the inactive precursor chem163S [4,18,19,20,21]. In individuals with obesity undergoing bariatric surgery, adipose tissue is a significant source of circulating chemerin, and much of it has been proteolytically processed [22].

Obesity is a significant contributor to insulin resistance and the subsequent type 2 diabetes mellitus (T2D) development [23,24,25]. A relationship has been described between circulating chemerin levels and BMI, with an increased production of chemerin potentially occurring in the liver [26,27]. There is a correlation between chemerin levels, both total and cleaved, and body mass index (BMI) in individuals with obesity and T2D [19,28,29]. Furthermore, chemerin has been proposed as a prognostic marker for T2D and may serve as a mediator of insulin resistance [30,31,32].

The correlation between BMI and chemerin levels is well researched [4,19,21,22,29,30,33,34,35]; however, the relationship between T2D and chemerin levels has not been well documented in either individuals with a normal weight and T2D (NWD) or individuals with obesity and T2D (OWD) [32,36]. There is an increased activation and subsequent proteolysis of chem163S in both humans and mice with obesity [22,34,35]. This exploratory study aimed to determine whether there was a difference in the concentration of different forms of chemerin between two groups of patients with T2D who had substantial differences in their BMI. Specifically, we investigated the relationship between cleaved and intact chemerin in NWD compared to their OWD counterparts. We hypothesized that the OWD group may exhibit elevated levels of cleaved chemerin due to a higher adiposity compared to the NWD group.

## 2. Materials and Methods

### 2.1. Acquisition of Samples

Human blood samples were obtained under protocols approved by the Stanford University Medical Center or Partners Healthcare Institutional Review Boards. Informed consent was obtained from the donors of plasma who had been recruited to two large NIH-funded randomized controlled clinical trials. Individuals with T2D with a body mass index (BMI) ranging from 25 kg/m^2^ to 70 kg/m^2^ (OWD) were randomized to the “Initiate and Maintain Physical Activity in Clinics” (IMPACT, 5R18DK096394) trial [37,38]. Individuals with T2D with a BMI below 25 kg/m^2^ (NWD) were randomized to the “Strength Training Regimen for Normal Weight Diabetics” (STRONG-D, 2R01DK081371) trial [39,40].

The study participants comprised adults between the ages of 18 and 80 who had been diagnosed with T2D, had hemoglobin A1c (HbA1c) levels between 6.5% and 13.0%, were not insulin-dependent, had a negative exercise stress test result, and had no precluding health issues. Smoking and a history of diabetes and its medical treatment were self-reported. Participants were recruited from the greater San Francisco Bay Area, California, from 2015 to 2019. For each study protocol, a three-arm randomized controlled trial (RCT) was conducted. Samples were drawn at the baseline screening visit of the participants who had consented to participate in the intervention and who underwent additional informed consent procedures before their participation in the Stanford Precision Health Biobank (Stanford GenePool). All the blood samples were collected in citrate, and the plasma was prepared by centrifugation before freezing at −80 °C for storage.

### 2.2. Preparation of Chemerin from Human Plasma

For the isolation of chemerin, the plasma was thawed, and 0.5 mL was mixed with 50 µL of heparin-agarose (Sigma) and 11 µL of ×50 Complete Protease Inhibitor (Roche Applied Science, Pleasanton, CA, USA). This mixture was shaken at 4 °C for 2 h before centrifugation to sediment the beads. After washing extensively with phosphate-buffered saline (PBS), the chemerin was eluted with PBS containing 0.8 M NaCl and the protease inhibitor cocktail.

### 2.3. ELISAs for Total Chemerin and chem163s

Since only 0.5 mL of plasma was available for these assays, we were unable to run all the ELISAs specific to different forms. Thus, we focused on the two key ones: total chemerin and chem163S. To measure the levels of total chemerin and chem163S in the plasma samples, 96-well ELISA plates were coated overnight at 4 °C with 4 µg/mL of a mouse monoclonal anti-human chemerin antibody (R&D Systems, Minneapolis, MN, USA) dissolved in PBS. The plates were blocked for 1 h with 1% BSA in PBS to prevent non-specific binding. Standards of recombinant total chemerin and chem163S were prepared as described in the previous section [22], serially diluted in 1% BSA in PBS for the calibration curve, and incubated for 2 h in the prepared wells. The samples were added to duplicate wells. After washing with 0.05% Tween 20 in PBS, the plates were incubated with 500 ng/mL of the relevant chemerin antibodies: either anti-chem163S prepared in-house [41] or anti-human chemerin (R&D Systems) in PBS with 1% BSA for 1 h. Following a further wash with PBS, the wells were incubated with either 100 ng/mL horseradish peroxidase (HRP)-conjugated goat anti-rabbit IgG antibody in the chem163S ELISA or 100 ng/mL HRP-conjugated streptavidin in the total chemerin ELISA (R&D Systems, Minneapolis, MN, USA) in 1% BSA in PBS for 1 h. After washing with 0.05% Tween 20 in PBS, the substrate, tetramethylbenzidine (Alpha Diagnostic International, San Antonio, TX, USA), was incubated for 10 min before terminating the reaction with a Stop Solution (Alpha Diagnostic International). Absorbance was measured at 450 nm, and the concentrations of both total human chemerin and chem163S were calculated from the calibration curves created with the purified standards.

### 2.4. Statistical Analysis

Continuous baseline characteristics were presented as the mean ± standard deviation. Categorical baseline characteristics were expressed as counts and percentages. Between-group differences for the baseline characteristics were assessed using an χ^2^ test or Fisher’s exact test (for small cell counts) for the categorical variables and a *t*-test (for normal distribution values) or a Mann–Whitney U test (for non-normal distribution values) for the continuous measures. For the primary outcome, in addition to total chemerin and chem163S levels, the level of cleaved chemerin (ng/mL) in each sample was calculated by subtracting chem163S (ng/mL) from the total chemerin (ng/mL). The concentrations of total chemerin, chem163S, and cleaved chemerin were compared between the groups using the two-tailed Student’s *t*-test with a significance value set to *p* < 0.05. To test the effect of HbA1c (%) on the concentration of each chemerin form (ng/mL), we used a linear regression analysis adjusted for the weight group effect. The group differences in the mean level of chemerin are hereby reported as the mean ± standard error of the mean (SEM). Effect sizes were also calculated, as standardized mean differences (SMDs), and interpreted using Cohen’s d thresholds—0.2 (small), 0.5 (medium), and 0.8 (large) [42]. Data analysis was performed from December 2023 to January 2024. All the analyses were performed using the R statistical software version 4.0.4 (R Project for Statistical Computing) [43], utilizing the R package ggplot v3.5.1 [44] for the figures. 

## 3. Results

### 3.1. Participant Characteristics

The demographic characteristics of the participants in this study are shown in Table 1. The age and levels of HbA1c were well matched between the groups (*p* > 0.05 for both comparisons), but, as expected, there was a significant difference in the BMI between the two groups (*p* < 0.001). In addition, the participants’ race was not balanced between the NWD and OWD groups.

### 3.2. Comparison of Total and Cleaved Chemerin Levels in NWD and OWD

The levels of chemerin (including total, intact chem163, and cleaved) were measured in the plasma samples from individuals categorized as NWD and OWD (Figure 1). We determined the levels of total chemerin and chem163S using ELISA in plasma samples obtained from both NWD and OWD participants, subsequently calculating the amount of cleaved chemerin by subtracting the value of chem163 from the total chemerin. Notably, there was no significant difference in the mean (± SEM) levels of total chemerin between the plasma samples from NWD (46 ± 3.4 ng/mL) and OWD (47 ± 1.9 ng/mL) (SMD = 0.07, *p* = 0.854). However, the mean (± SEM) levels of intact chem163S were significantly higher in the plasma samples collected from OWD (30 ± 1.3 ng/mL) compared to those from NWD (20 ± 1.3 ng/mL) (SMD = 1.39, *p* < 0.0001). Furthermore, we observed that the mean (± SEM) levels of cleaved chemerin were significantly lower in the plasma samples obtained from OWD (17 ± 1.8 ng/mL) compared to those from NWD (27 ± 2.9 ng/mL) (SMD = 0.80, *p* = 0.011). The SMDs when comparing both chem163S and cleaved chemerin between NWD and OWD were both considered large.

### 3.3. Correlation of Chemerin Levels with HbA1c in NWD and OWD

We analyzed whether there was any correlation between the levels of each chemerin form (including total, intact chem163, and cleaved) in the plasma samples and HbA1c (%) in the individuals categorized as NWD and OWD. Across the range of HbA1c (%), we observed no correlation between the HbA1c (%) levels and the chemerin levels in OWD and NWD, suggesting that glucose homeostasis is not related to chemerin levels in these individuals with T2D (*p* > 0.05 for all, Figure 2).

## 4. Discussion

In this study investigating the difference in the concentrations of varying forms of chemerin among individuals with T2D but with different BMIs, we found that the total chemerin levels were similar in the NWD and OWD groups. This suggests that T2D equalizes the chemerin levels, irrespective of obesity. However, the plasma samples from OWD had significantly lower cleaved chemerin levels than those from NWD, indicative of higher chemerin processing in the NWD cohort and, possibly, a reduced inflammatory state in OWD compared to NWD. Finally, we found no association between chemerin forms and % HbA1c. Our findings indicate that the plasma samples collected from individuals with a normal weight and type 2 diabetes exhibited elevated proteolytic and inflammatory activity compared to their T2D counterparts with obesity. Our original hypothesis was that individuals who were part of the OWD group would have higher levels of chemerin cleavage due to the interaction between T2D and obesity. Instead, we found the opposite result: there was more chemerin cleavage in NWD than in OWD.

All the individuals in this study exhibited more chemerin processing than the control individuals [22,41]. Individuals with a normal weight and T2D exhibited enhanced chemerin processing, suggesting that augmented levels of proteolytic activity may be attributed to increased systemic inflammation. This supports the notion that the chemerin system’s activity may predominantly occur in tissues subject to active inflammation, such as adipose tissue in people with obesity [19]. Despite similar levels of total chemerin between the groups, the OWD group had significantly greater levels of intact chem163S (*p* < 0.001) and significantly reduced levels of cleaved chemerin (*p* = 0.011) compared to NWD.

The disparities observed between NWD and OWD might be rooted in two distinct factors, as our study primarily explored correlations to infer causality and associations among these variables. The observed alterations in the chemerin levels could be a result of differences between the composition of the patient groups, the social and economic conditions under which they live, the length of time since the disease first manifested, or a combination of all these factors, all of which have the potential to be modulated by additional unidentified factors.

In Saudi women with matched BMIs, chemerin levels were higher in women with T2D [30]. In a second study involving both sexes, the only groups with T2D that exhibited higher levels of chemerin than the subjects with normal BMIs were those with either obesity or severe obesity [45]. The data in this study show that individuals with T2D who differ in terms of their BMI had different levels of chemerin processing (Figure 1); however, the HbA1c levels did not influence the levels of chemerin processing (Figure 2). In contrast, our companion study showed that the levels of plasma glucose correlated with both the levels of chemerin and the amount of its cleavage, but that BMI was a very significant confounding factor (Zhao et al., *Biomedicines* in press) [46]. Taking these two studies together suggests that the effects of both BMI and T2D on chemerin levels are not independent of each other but, instead, take place via a common mediator. It should be noted that, while these studies show a correlation, they do not demonstrate whether chemerin is upstream or downstream from the joint effects of BMI and T2D.

The mechanisms underlying these observations remain unclear in both animal and human studies, with many not distinguishing between the effects of obesity and T2D [47,48]. In mice, chemerin regulates pancreatic beta-cell function, with a lack of chemerin causing reduced glucose-dependent insulin secretion [49]. Nonetheless, studies have associated an increase in chemerin levels with elevated instances of myocardial infarction (MI) and stroke [7,50,51]. Elevated proteolytic activity in this group suggests more inflammation and, consequently, an increase in mortality risk. Our findings revealed more cleavage of chemerin in individuals with a normal weight with T2D compared to individuals with obesity and T2D, but the chemerin remained predominantly intact, particularly within the group with a high BMI. One hypothesis is that chemerin levels would be elevated in individuals with obesity and T2D versus individuals with a normal weight and T2D. However, our research found no difference in the levels of total chemerin between the individuals with a normal weight and T2D and the individuals with obesity and T2D, but the weight increase was more marked in individuals with diabetes than in individuals without diabetes.

Chemerin is a critical player in metabolic and immune regulation, particularly in diseases which provoke systemic inflammatory conditions [8,52,53,54]. The proteolytic cleavage and regulatory patterns of chemerin are consistent across human and mouse models, with several studies indicating that variation in inflammatory responses between conditions such as rheumatoid arthritis and osteoarthritis is linked to the generation of different chemerin forms by proteolysis [11,12].

The homology between chemerin in mice and humans, as well as its receptors, chem1 and chem2, suggests that data from mouse disease models will be generally applicable to humans. Obesity, characterized by increased adiposity, has been associated with the induction of inflammation in adipose tissues, leading to metabolic syndromes and the impairment of insulin sensitivity in major tissues [55,56,57]. Chemerin is positively correlated with body weight and metabolic syndrome markers [19,21]. In mice, chemerin plays a key role in the development of both brown and white adipose tissue (BAT and WAT), and chemerin deficiency leads to the dysregulation of the glucose metabolism, increases in weight, and insulin resistance [16,49,58]. The presence of active BAT has been shown to be more prevalent in lean mice, implicating BAT in energy storage and metabolisms concerning obesity. Chemerin’s role in glucose uptake in white adipose tissue (WAT) and its involvement in modulating glucose homeostasis underscore its varied functions in different types of adipocytes [4,59].

In T2D, chemerin is higher in the blood of individuals with concurrent hypertension, although its levels may not vary significantly from those in individuals with normal glycemia [60]. The relationship between chemerin and the glucose metabolism is further complicated by findings from a recent exercise intervention, where a decrease in serum chemerin levels was associated with improved insulin sensitivity [61].

There were some limitations to our study. First, this study was limited by the small number of participants enrolled. Additionally, reliance on self-reported type 2 diabetes (T2D) status in the original clinical trials posed challenges. Several participants were unaware of their T2D diagnosis until the baseline HbA1c screening, which hindered our determining of the time passed since the onset of T2D. Moreover, race and ethnicity were not well matched, which may be a confounder factor (Figure 1). It should be noted that over 70% of individuals in the NWD group identified as Asian, compared to less than 25% in the OWD group. Individuals of Asian ethnicity tend to develop T2D at much lower BMI levels than those of other ethnic groups, possibly due to increased visceral adiposity [62]. Therefore, further investigations into the relationship between these chemerin forms and weight status should involve other ethnic groups and incorporate a larger cohort of participants. Finally, there was no normoglycemic control group.

This study also had several strengths. One major strength of this study was that it was the first investigation into how diabetes and obesity concurrently influence chemerin levels rather than examining each condition’s link to chemerin in isolation. Specifically, our study compared the chemerin levels in individuals with a normal weight who had type 2 diabetes against those in individuals with obesity. Our findings indicate that diabetes and obesity, together, primarily contribute to the variations in chemerin levels in patients with T2D. The use of two distinct cohorts—individuals with a normal weight and T2D and individuals with obesity and T2D—was instrumental in clarifying the intertwined roles of these conditions in affecting chemerin’s levels and processing.

Exploring chemerin’s function in individuals with T2D but a normal BMI highlights its role as a link between the development of obesity and T2D. The complexity of chemerin’s functions, as highlighted by the heterogeneity of its forms and its potential to precede T2D onset, warrants further investigation. Future research should focus on the specific interactions of chemerin in individuals with a normal weight and T2D, the secretion dynamics of chemerin relative to other adipokines, and whether isoform-specific expression can discriminate between T2D in individuals with a normal weight and those with obesity. The continuation of this research trajectory is essential to substantiate chemerin’s potential as a biomarker in clinical settings. Future studies should also examine the specific relationships between chemerin, obesity, and type 2 diabetes using more precise measures of obesity, such as DEXA (dual X-ray absorptiometry) scans [63].

## 5. Conclusions

Our research suggests that there is more extensive proteolytic cleavage of chemerin in the plasma of individuals with a normal weight and type 2 diabetes. Given the lack of an association between the level of chemerin and HbA1c in individuals with a normal weight and T2D and individuals with obesity and T2D, future studies should consider implementing diagnostic modalities for insulin resistance beyond traditional markers. Furthermore, future studies should aim to increase the sample size and consider additional variables such as age, sex, and medication history. Additionally, such studies should include an exploration of the levels of specific chemerin forms to delineate chemerin activity more accurately in individuals with a normal weight and T2D compared to those with T2D and obesity.

## Figures and Tables

**Figure 1 biomedicines-12-00983-f001:**
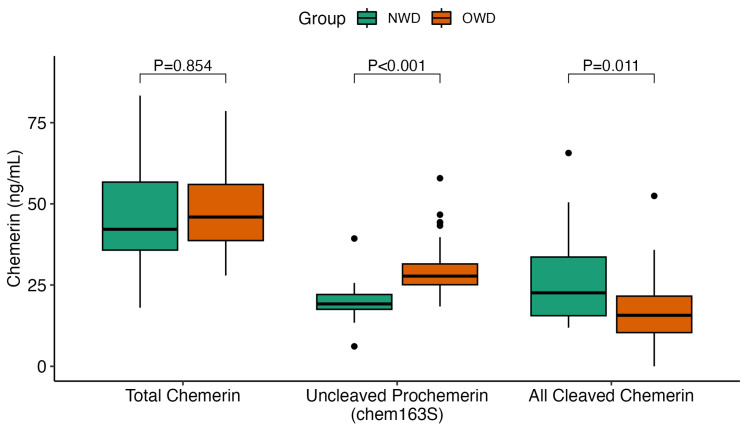
Box–whisker plot (displaying the 90/10 percentile at the whiskers, the 75/25 percentiles at the boxes, and the median in the center line) of each of the chemerin forms from the samples collected from OWD versus NWD. A Student’s *t*-test was conducted to compare the levels of each chemerin form between the groups.

**Figure 2 biomedicines-12-00983-f002:**
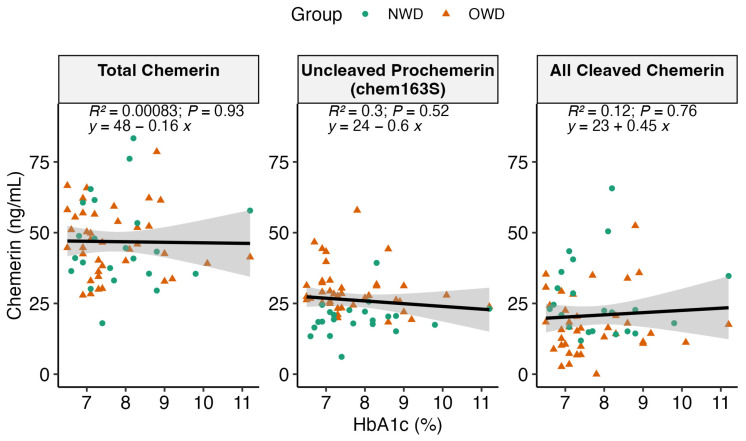
Regression analysis revealing no significant association between HbA1c (%) and the concentration of each chemerin form (ng/mL). (*p* > 0.05). The brown triangles denote the samples collected from OWD, and the green circles denote the samples collected from NWD.

**Table 1 biomedicines-12-00983-t001:** Participants’ characteristics at the baseline.

		NWD (N = 22)	OWD (N = 39)	*p*-Value ^1^
Age (years), mean ± SD ^2^		58.5 ± 10.44	57.7 ± 10.4	0.763
Sex, N (%)				
	Female	11 (50.0%)	14 (35.9%)	0.421
	Male	11 (50.0%)	25 (64.1%)	
Race/Ethnicity, N (%)				0.002
	African American/Black	1 (4.5%)	3 (7.7%)	
	Asian	16 (72.7%)	9 (23.1%)	
	Hispanic or Latino	0 (0.0%)	6 (15.4%)	
	Non-Hispanic White	5 (22.7%)	18 (46.2%)	
	Other Race	0 (0.0%)	3 (7.7%)	
Body Mass Index [BMI] (kg), mean ± SD		23.8 ± 1.41	36.2 ± 7.13	<0.0001
Hemoglobin A1c [HbA1c] (%), mean ± SD		7.87 ± 1.11	7.69 ± 1.05	0.533
Duration of Diabetes (years), mean ± SD ^3^		11.1 ± 8.89	6.06 ± 5.70	0.0324
Smoking Status ^3^				0.876
	Never	16 (72.7%)	27 (69.2%)	
	Past Smoker	3 (13.6%)	8 (20.5%)	
	Current Smoker	1 (4.5%)	2 (5.1%)	

^1^ χ^2^ test or Fisher’s exact test (for small cell counts) have been used for the categorical variables, while a *t*-test (for normal distribution values) or Mann–Whitney U test (for non-normal distribution values) have been used for the continuous measures to examine the distribution of the demographic characteristics of the participants by weight group at the baseline. ^2^ SD = standard deviation. ^3^ Self-reported measures. Eleven data points missing/unknown/declined to state for the self-reported duration of diabetes, and four data points missing/unknown/declined to state for the smoking status.

## Data Availability

Data are available upon application to the corresponding authors. The data are not publicly available due to privacy concerns.

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
