# Peer review of "Chemerin Levels in Individuals with Type 2 Diabetes and a Normal Weight versus Individuals with Type 2 Diabetes and Obesity: An Observational, Cross-Sectional Study"

_biomedicines, 2024, doi:10.3390/biomedicines12050983_

Round 1

Reviewer 1 Report (Previous Reviewer 2)

Comments and Suggestions for Authors

I reviewed the initial submission.  The authors have addressed each of my concerns adequately.  No further comments/suggestions. 

Author Response

The authors thank the reviewer for their suggestions that improved our manuscript.

Reviewer 2 Report (New Reviewer)

Comments and Suggestions for Authors

·      The adjective obese should be removed since it is stigmatizing, in alternative can be substituted by “with obesity”

·      The nature of the study should be mentioned in the title

·      The keywords should be arranged in alphabetical order. Moreover include only keyword that does not appear in the title. 

·      After the aim of the study in the introduction section, a hypothesis should be mentioned

·      A power analysis is needed to justify the sample size.

·      Do body composition variable available) body fat?

·      The discussion should be better restructured, I advise authors to adhere to the following order:

-       Main findings of the study and their comparison with similar previous published studies

-       The clinical implication of the findings

-       The strengths and limitations

-       The new directions for future research

·      The article need to be enriched in references on the topic

Round 2

Reviewer 2 Report (New Reviewer)

Comments and Suggestions for Authors

Non. 

This manuscript is a resubmission of an earlier submission. The following is a list of the peer review reports and author responses from that submission.

Round 1

Reviewer 1 Report

Comments and Suggestions for Authors

In this study, authors showed the total form of chemerin levels in the plasma were similar in the normal weight T2DM and overweight T2DM groups. And cleaved form of chemerin levels in the plasma were decreased in overweight T2DM group compared to normal weight T2DM group. In addition, there was no correlation between chemerin and HbA1c in the both type of T2DM gourps. It is difficult to determine the importance of chemerin in diabetes through these results.

Comments on the Quality of English Language

Minor editing of English language required

Reviewer 2 Report

Comments and Suggestions for Authors

In this study, the authors describe the relationship between the adipokine chemerin and body mass in type II diabetic patients. In short, they report no difference in total chemerin level, but there are significant differences in the level of cleaved versus uncleaved chemerin among the groups. Proteolytic activation/deactivation of chemerin has previously been described. The authors found no relationship between measures of chemerin and %Hb1ac which was used as a proxy of glucose metabolism.  However, the authors refer to a companion paper that demonstrated a relationship between chemerin and glucose metabolism using another measure. It is unclear why these data were not included here.

As noted by the authors, a significant shortcoming of the study is the relatively small sample size and the poor matching between groups regarding race and ethnicity. More significantly, there are no data from normal weight individuals, as well as overweight individuals, that are not diabetic. This is not addressed. Also, why was the relationship between the degree of adiposity and chemerin level beyond overweight versus normal weight considered?  

The discussion is much too lengthy and includes a lot of supposition concerning the underlying cause of the observed differences in chemerin levels among the groups. For example, it is stated there are differences in the inflammatory state among the study groups, but no measures of inflammation are reported. Despite the well-known association between inflammation, obesity and T2D, it can not be assumed that those tenets hold true in the study populations. The sections discussing cardiovascular disease are not relevant here because no measures of CV function were included.  

Reviewer 3 Report

Comments and Suggestions for Authors

-          This is a study analysing the relationship between total chemerin and cleaved chemerin levels and BMI in a group of patients with T2DM. The authors found similar levels of total chemerin in normal weight versus obese diabetic patients and lower levels of cleaved (active) chemerin in the obese group. This are quite surprising results, that contradict previous studies on this topic and even the author’s initial hypothesis. Since there are few research on this subject, this study might prove interesting, but I believe there are certain concerns:

-          First, the number of patients is quite low, so it is difficult to draw a significant conclusion.

-          Second, there are important differences in ethnicity between the two groups – more than 70% in NWD group are Asians, compared to less than 25% in OWD group. It is well known that people with Asian ethnicity develop diabetes at much lower BMI-s than other ethnic groups, probably due to increased visceral adiposity. I believe this is a major fault of this study. Also, there is no information regarding diabetes history, other comorbidities or associated pathologies, smoking status or other factors that might influence chemerin levels

-          Tha lack of a normal weight non-diabetic group is also a significant issue